A collaborative data storage with incentive mechanism for blockchain-based IoV

Shi Quan 1
Wang Lankai 1
Chen Chen chenchen86@ntu.edu.cn 2
1 School of Transportation and Civil Engineering, Nantong University , Nantong , China
2 School of Information Science and Technology, Nantong University , Nantong , China
Schifanella Rossano
Electronic publication date: 2025 May 27
Publication date: 2025
Volume: 11
Electronic Location ID: e2849
Received 2024 Nov 14; Accepted 2025 Mar 31
Copyright: ©2025 Shi et al.
Copyright year: 2025
Copyright holder: Shi et al.
License: This is an open access article distributed under the terms of the Creative Commons Attribution License, which permits unrestricted use, distribution, reproduction and adaptation in any medium and for any purpose provided that it is properly attributed. For attribution, the original author(s), title, publication source (PeerJ Computer Science) and either DOI or URL of the article must be cited.
License URL: https://creativecommons.org/licenses/by/4.0/

Keywords: Collaborative data storage, Incentive mechanism, IoV, Blockchain

Funding: National Natural Science Foundation of China 62476145 Humanity and Social Science Foundation of Ministry of Education of China 24YJAZH126 Technology Research and Development Talent Foundation of Jiangsu Province Transportation Technology and Achievement Transformation Foundation of Jiangsu Province 2024G01 Postgraduate Research & Practice Innovation Program of Jiangsu Province SJCX24_2013 This work was supported by the National Natural Science Foundation of China (62476145), the Humanity and Social Science Foundation of Ministry of Education of China (24YJAZH126), the 6th “333 Talents” Technology Research and Development Talent Foundation of Jiangsu Province, the Transportation Technology and Achievement Transformation Foundation of Jiangsu Province (2024G01), and the Postgraduate Research & Practice Innovation Program of Jiangsu Province (SJCX24_2013). The funders had no role in study design, data collection and analysis, decision to publish, or preparation of the manuscript.

==============================
As the volume of data in the Internet of Vehicles (IoV) continues to grow, challenges such as insufficient storage capacity and potential privacy breaches become more pronounced. To address these issues, this article proposes a novel collaborative data storage scheme with an incentivization mechanism, termed Blockchain-Based Collaborative Data Storage with Incentive Mechanism for IoV (CDS-BIoV). The CDS-BIoV framework consists of vehicles, roadside units (RSUs), and cloud infrastructure. In the first phase, vehicles collect and transmit data to their nearest RSU nodes. To encourage active participation in data reception and storage, an incentive mechanism is introduced to motivate RSU nodes. Two algorithms are developed: the Incentive Mechanism Collaborative Data Storage Algorithm (I-CDSA) and the Data Offloading Algorithm (DOA). The I-CDSA uses a competitiveness matrix to incentivize RSU nodes to minimize storage consumption, while the DOA employs incentives to secure additional cloud storage for offloading data. Experimental results show that the CDS-BIoV scheme reduces storage consumption by up to 93% compared to the Generic Parallel Database (GPDB), particularly as the number of blocks increases, effectively alleviating storage capacity limitations.

Introduction

With global motor vehicle registrations projected to exceed 2 billion in the coming decade due to rapid industrialization and urbanization (Jia et al., 2015), traffic congestion and road safety concerns are becoming increasingly critical. Intelligent transportation systems (ITS), particularly the Internet of Vehicles (IoV), play a vital role in mitigating these challenges by enhancing real-time traffic management and safety through vehicle-to-vehicle (V2V) and vehicle-to-infrastructure (V2I) communication (Aman et al., 2024). By leveraging networks such as 3G, 4G, 5G, Wi-Fi, and Bluetooth, IoV enables seamless data sharing among vehicles and roadside units (RSUs), improving mobility and reducing accidents (Wang et al., 2022a). However, the explosive growth in vehicular data volume introduces severe challenges related to data storage, processing efficiency, and security (Kumar et al., 2023).

Traditional IoV architectures rely on centralized data storage managed via cloud platforms, which struggle to handle high data throughput and real-time processing demands (Franzese, Zhang & Mahmoud, 2010). This centralized structure presents multiple vulnerabilities, including single points of failure, data tampering risks (Zhang et al., 2022), and unauthorized access threats (Wang et al., 2021). Additionally, prolonged system operations and hardware fatigue can lead to infrastructure failures, jeopardizing the integrity and availability of critical traffic data (Liu et al., 2022a). Given the high-stakes nature of IoV applications, ensuring robust, scalable, and tamper-resistant data storage is essential (Wang et al., 2022b).

Blockchain technology (Nakamoto, 2008) offers promising solutions to these challenges by enabling decentralized, secure, and transparent data management (Ma, Wang & Zhao, 2020). Through cryptographic mechanisms and consensus protocols, blockchain prevents unauthorized data modifications and eliminates reliance on third-party validation (Shao et al., 2018). However, the direct implementation of blockchain in IoV systems faces scalability concerns (Vishwakarma & Das, 2022), as RSUs possess limited computational power and storage capacity, restricting their ability to handle extensive traffic data efficiently (Tong et al., 2023).

To address these issues, this article proposes a novel Collaborative Data-Storage Blockchain for IoV (CDS-BIoV), incorporating an incentive-driven approach to optimize storage distribution. Inspired by Yu et al. (2022), our model employs the entropy weight method to enhance storage allocation across RSUs. Instead of burdening all nodes with full data replication, data are selectively stored on RSU nodes, reducing redundancy while maintaining accessibility. The RSU storage incentive mechanism (RSIM) encourages participation by rewarding RSUs for data reception and storage contributions. These transactions are implemented via Ethereum smart contracts, with earned incentives facilitating cloud-based data offloading when necessary (Sharma, Jindal & Borah, 2021). This approach enhances both the security and scalability of IoV storage systems while maintaining efficient real-time data access.

The primary contributions of the system proposed in this article are as follows:

(1) This study establishes a decentralized data collaborative storage model utilizing blockchain technology to achieve decentralized data storage and collaborative management. This model enhances data security and reliability by distributing data across multiple RSU nodes, thereby preventing single points of failure and centralized risks.

(2) A data collaborative optimization storage algorithm based on RSUs was designed to efficiently distribute and store data across multiple RSU nodes. This algorithm enhances both the speed and efficiency of data storage, while also improving the system’s fault tolerance and stability, thereby ensuring data reliability and continuity in dynamic vehicular environments. In addition, through collaborative optimization, the algorithm can adapt to real-time data changes, providing a flexible and efficient storage solution.

(3) The study designs an incentive mechanism and a data offloading algorithm within a decentralized data collaborative storage scheme. The incentive mechanism encourages more nodes to participate in sharing storage and computing resources, thereby enhancing system resource utilization and data processing efficiency. The data offloading algorithm allocates part of the storage tasks to the cloud, reducing the burden on local nodes and improving the system fault tolerance and stability, ultimately achieving higher storage reliability, faster response times, and flexible resource allocation.

(4) This article employs smart contracts to materialize the incentives outlined in this study, deploying them within the Ethereum network. The experimental results demonstrate that this approach not only improves data storage efficiency but also effectively prevents the formation of selfish nodes in the network.

The remainder of this article is organized as follows. ‘Related Work’ provides an overview of related research. ‘Materials & Methods’ outlines the model and principles of the proposed solution. ‘Experimental Evaluation’ details the implementation of the solution. ‘Discussion’ evaluates the system performance through experiments. Finally, the outcomes of the study are summarized and potential directions for future research are discussed.

Related Work

This section explores various blockchain-based storage optimization techniques for the Internet of Vehicles (IoV), including node-dependent optimized storage, encoding-based storage, and cloud-based solutions. These approaches aim to address issues like storage efficiency, scalability, and performance overhead. Additionally, incentive mechanisms for enhancing security and privacy in IoV are discussed, focusing on reputation systems and adaptive payment schemes to promote data sharing while safeguarding user privacy. These advancements contribute to more efficient, secure, and privacy-preserving IoV systems.

Storage optimization techniques for blockchain

In the realm of blockchain storage optimization techniques, three primary approaches are explored: node-dependent optimized storage (Jayabalan & Jeyanthi, 2022), incorporating coding mechanisms (Han et al., 2024), and utilizing cloud storage for backup and offloading (Fu et al., 2024). We survey the recent literature and present a summary, as shown in Table 1.

Table 1 The summary of literatures.

Scheme	Concept applied	Major contributions	Drawbacks	
Zhang et al. (2022)	A hierarchical edge-cloud blockchain (LayerChain)	A hierarchical structure is used to optimize storage and dissemination efficiency.	At present, the scale of simulation experiments is small, and the performance of the system needs to be further verified when it is applied	
Sohan et al. (2021)	IPFS	The dual blockchain approach works by adding a reference to the master block to the ledger.	The issue of vehicle complexity and security is not considered	
Yahaya et al. (2022)	A double blockchain	1. An incentive mechanism and a reputation system to reduce the selfish and distrustful behavior of IoV nodes. 2. A hybrid of Rivest Shamir Adleman and Advanced Encryption Standard-256 (AES-256+RSA) to protect the proposed system from passive and active attacks.	The system has high complexity and high performance overhead.	
Spataru, Pungila & Radovancovici (2021)	A semantically-different blockchain	1. A semantically-different blockchain architecture, which introduces a new type of nodes whose purpose is to enhance the storage. 2. A highly efficient storage model combined with a hybrid compression mechanism by smart contracts.	It remains to be verified that this model can be used in the Internet of Vehicles.	
Li et al. (2021)	Lightweight blockchain	1. A blockchain storage optimization scheme based on RS erasure code. 2. An improved PBFT blockchain consensus mechanism based on reward and punishment strategy.	The dynamic joining and leaving of vehicles must be considered, and the efficiency of the model should be improved to meet the requirements of vehicle networks.	
Tiwari & Lalitha (2021)	Secure Raptor Encoder and Decoder (S-RED)	The method uses Raptor codes as a basis and develops S-RED with counter packet-decoding loss, as well as an error-resilient double-layer peer offsetting model.	S-RED requires strict decoding conditions, which are not easy to achieve, and the failure rate of decoding may be high in the case of packet loss.	
Raman & Varshney (2021)	Dynamic distributed storage method	Secret key sharing and private key encryption technology are used to encode blockchain data, ensuring the security of data storage	The increase of memory and the cost of redundant storage may increase storage and repair costs.	
Yin et al. (2021)	Intelligent vehicle data storage system	1. The system manages the dynamic data storage of vehicles and the storage of passing vehicles and archive areas. 2. The storage area of dynamic vehicles is created and stored in a distributed manner.	The impact of the dynamic growth of data and the redundancy mechanism must be considered to ensure reliability.	
Liao et al. (2022)	A graph-partitioned blockchain supported by DAG-based storage	1. A graph-partitioned blockchain that uses a transaction graph supported by DAG-based storage. 2. DAG blocks are used to create a transaction graph and classify transactions based on transaction characteristics of asynchronous data storage.	The influence of node storage growth rate and the redundancy mechanism are not fully considered.	
Samy et al. (2022)	Blockchain and deep reinforcement learning	An integrated model of resource allocation and task offloading are constructed to optimize energy and time costs.	The optimization model assumes a small number of users, and it is unclear if large-scale scenarios can be implemented. Additionally, model complexity may be unbearable.	

Node-dependent optimized storage

In the study (Sohan et al., 2021), the authors enhanced throughput by employing a dual blockchain approach and alleviated the storage scarcity issue by integrating an InterPlanetary File System (IPFS). Building upon Sohan et al. (2021)’s work, Yahaya et al. (2022) applied the dual blockchain method in an in-vehicle system to optimize IoV data storage, employing encryption technology to safeguard the system against potential attacks. These two studies adopted dual blockchain in conjunction with IPFS and IoV to reduce storage consumption. To address the storage requirements of smart contracts, the study proposed a continuously adaptive approach that enhances the storage efficiency of smart contracts by introducing a new type of node (Spataru, Pungila & Radovancovici, 2021). This novel node minimizes the storage requirements for smart contract execution and transmission, thereby boosting the overall blockchain throughput without compromising overall blockchain performance. However, increasing the number of blockchain and introducing new node types necessitates additional deployment costs, suggesting that further advancements are essential.

Encoding optimized storage

Li et al. (2021) tackled the storage overhead of blockchain nodes and the associated storage burden by proposing a blockchain storage optimization scheme based on the Reed-Solomon (RS) erasure code. Similarly, Tiwari & Lalitha (2021) partitioned blockchain nodes into segments and introduced a security codec based on Raptor coding to address the issue of excessive storage consumption in full nodes. Their approach reduces storage consumption while maintaining a constant decoding time relative to the number of input blocks (Tiwari & Lalitha, 2021). To counteract the increased storage cost of blockchains in large networks with high transaction rates, Raman & Varshney (2021) employed distributed storage via private key encryption to design an encoding scheme that diminishes node storage costs by storing a portion of each transaction at each node. A dynamic partitioning algorithm was also utilized to enhance transaction data integrity. However, the encoding process may increase the encoding consumption of the system, potentially affecting the overall performance.

Optimized storage in the cloud

Vehicle information and communication security have garnered increasing attention in the Internet of Vehicles (IoV) domain (Laghari et al., 2023). Addressing the challenges of substantial data update overhead and inadequate consensus algorithm intelligence in traditional blockchain, Yin et al. (2021) proposed a blockchain data storage system that supports data updates. The system incorporates multiple replica storage for preserving data sources and stores the data source address within the blockchain. By partitioning data and employing smart contracts, the system reduces the same data upload size, thereby minimizing the system overhead. In addition to the aforementioned issues, traditional blockchain networks face significant challenges related to storage costs and transaction throughput. Liao et al. (2022) further optimized LayerChain by integrating the storage strategy of a generic parallel database (GpDB) and a graph partitioning algorithm based on transaction freshness. This approach reduces the storage costs of edge servers by offloading data to cloud services. However, utilizing third-party data backup and offloading may increase network throughput consumption, potentially affecting network performance. Samy et al. (2022) employed a comparative reinforcement learning approach to derive near-optimal task offloading decisions, offloading data from the physical layer to the cloud, and conserving system resources. Ren et al. (2024) designed a blockchain-based data storage architecture. To reduce the coherence delay, this article proposes a reputation-aided practical Byzantine fault tolerance algorithm for reduced grouping reduction, RGS, and a data query algorithm (Ren et al., 2024). By introducing the LevelDB database, a transaction index chain coupled with a vehicle ID is established in the blockchain. This method avoids time-consuming block traversal during a data query.

The above schemes propose various blockchain storage optimization methods, but they also face issues such as system scalability and high-performance overhead. This study addresses these issues by developing a blockchain-based data storage model for the Internet of Vehicles (IoV). It employs an incentive mechanism to encourage roadside units to collaborate on data storage optimization, thereby enhancing the efficiency of blockchain-based IoV data storage. Additionally, smart contracts are deployed and simulated in Ethereum.

Enhancing security and privacy in biov through incentive mechanisms

In the Internet of Vehicles (IoV), ensuring the security and reliability of data storage is crucial due to the vast amounts of sensitive and real-time information exchanged among vehicles (Liu et al., 2022a; Liu et al., 2022b), RSUs, and central servers. Traditional IoV data management approaches often suffer from centralization risks, including single points of failure, data tampering, and unauthorized access (Ma et al., 2022). To address these challenges, blockchain technology has emerged as a decentralized solution that enhances data security, integrity, and traceability (Hizal, 2023).

Security

To tackle the challenges of information security and management within traditional in-vehicle self-organizing networks, Firdaus & Rhee (2021) leveraged federated chains for secure data storage, thereby safeguarding against system failures. Researchers incentivize vehicle participation through reward mechanisms, which not only ensure data authenticity but also enhance the overall performance of the IoV. Khalid et al. (2021) proposed the integration of blockchain technology on RSUs to capture and store vehicle transmission information via the Interplanetary File System (IPFS), addressing the issues of opacity and insecurity in traditional IoV data management. They introduced a reputation system to assess the accuracy of vehicle-transmitted data, offering monetary rewards to vehicles that provide correct information. In response to concerns about message reliability, Firdaus, Rahmadika & Rhee (2021) introduced a decentralized, trusted data-sharing management system for IoVEC. This system employs a federated chain to verify message trustworthiness through neighboring vehicles and rewards honest participants with an incentive mechanism. Yahaya et al. (2022) contribute to the study of incentive mechanisms by proposing reputation systems and incentives within dual blockchain in-vehicle systems. These mechanisms aim to mitigate the presence of selfish nodes and handle suspicious behaviors in vehicular networks, thereby promoting a more trustworthy and cooperative network environment.

Privacy

To enhance the credibility of messages transmitted by vehicles within the IoV ecosystem, Firdaus, Rahmadika & Rhee (2021) proposed a reputation rating system. In this system, neighboring vehicles are involved in the verification process of the sender’s information, and the system rewards these adjacent vehicles based on the verification outcomes, thereby incentivizing honest participation and ensuring the quality of shared data (Firdaus, Rahmadika & Rhee, 2021). To safeguard user privacy during information sharing and encourage users to exchange data with RSUs in the IoV, Alam et al. (2021) introduced an adaptive neuro-fuzzy payment scheme for blockchain-based traffic communication. This innovative scheme not only motivates user participation in data sharing but also maintains the confidentiality of their personal information, striking a balance between participation incentives and privacy protection.

In summary, researchers have designed incentive mechanisms by introducing reputation systems and adaptive payment schemes to enhance the security, privacy protection, and fairness of blockchain-based Intelligent Internet of Vehicles (BIoV). However, this increases the complexity of the system. This study considers the communication capabilities, storage capacities, and historical behavior records of nodes and proposes an incentive mechanism based on collaborative storage and offloading to improve the fairness and security of participation.

Materials & Methods

This study presents a hierarchical data storage model for the BIoV, consisting of four layers: perception, edge, blockchain, and cloud. The model utilizes an incentive mechanism for collaborative data storage, where RSUs assess their competitiveness and offload data to cloud storage when necessary. By integrating a competition matrix and smart contracts, the system ensures efficient data storage while incentivizing RSUs to maintain network cooperation. The approach enhances system robustness, reduces storage redundancy, and optimizes overall storage efficiency.

The system model

Adhering to a hierarchical approach, this article presents a data storage model for BIoV (Mathur et al., 2023), as depicted in Fig. 1. The model comprises a four-tier data storage framework: perception, edge, blockchain, and cloud layers. Each layer is detailed as follows:

Figure 1 Data storage framework.

A four-layer blockchain-based data storage model for the Internet of Vehicles (BIoV), including the perception layer, edge layer, blockchain layer, and cloud layer. The perception layer facilitates information exchange between vehicles and RSUs. Edge layer RSUs handle data processing and consensus, competing for data block storage through smart contracts to receive incentives. The blockchain layer selects optimal RSUs for collaborative storage to reduce storage costs. The cloud layer manages low-frequency data offloading, ensuring data redundancy and preventing data loss.

Perception layer: the primary function of the perception layer is to facilitate communication between vehicles and RSUs regarding road information. This framework assumes that all information, such as V2X data, is trustworthy. To ensure the legitimacy of vehicles within the IoV network, all vehicles must be authorized by a trusted party (TP) edge layer: in this framework, the edge layer refers to the RSUs, which are responsible for communicating with vehicles within a specific range, aggregating messages, implementing consensus mechanisms, generating data blocks, storing data, and offloading data. The blockchain network is deployed on RSU nodes using the proof-of-work (PoW) consensus algorithm. RSUs compete to store data blocks and earn incentives through smart contracts.

Cloud layer: this layer consists of cloud storage that manages data offloading from RSUs based on block access frequency. It secures multiple cloud storage spaces to house infrequently accessed data offloaded by RSUs, thereby ensuring data redundancy. This approach also safeguards against data loss owing to cloud server failures or corruption.

Blockchain layer: in the framework, the blockchain service layer is responsible for selecting RSUs nodes and calculating incentives. It selects RSUs nodes based on a competitiveness matrix calculation, with the results transmitted to the RSUs within the edge layer to facilitate cooperative data-block storage, thereby reducing the storage overhead of the blockchain network system. Furthermore, the blockchain service layer’s incentive mechanism calculates the number of incentives sent by the sender RSUs to each receiver RSUs, as well as the value of the internal incentives spent on data offloading and transmission. Once the incentive calculations are completed, the acquisition and expenditure of incentives are executed via smart contracts.

Within the system, vehicles in the perception layer aggregate the gathered information into data blocks and transmit these to the RSUs nodes in the edge layer within their range. After calculating via the blockchain service layer, the RSU node selects the RSUs node that will receive the data block and provides incentives to the recipient. When an RSU node requires data offloading, it must acquire cloud storage space for this purpose. The meanings represented by the symbols used in the text are shown in Table 2.

Table 2 Explanation of symbols.

Symbols	Explanation	
DR i	The driving record.	
L i	The path loss of the distance between node i and the sending node.	
Mu i	The storage utilization.	
M i	The remaining storage size in the node after the offloading process.	
γ n	The weighting factor of indicator n.	
Ci ^	The initial remaining storage size before offloading.	
r∗Mi,max	An integer multiple of the data block size unloaded from the node.	
Pi′	The initial incentive value.	
P i	The incentive of RSU node i.	
Mo i	The remaining amount of incentive.	
Mo¯i	The original amount of incentive.	
Sdb i	The spending incentive by the node to broadcast data blocks.	
P¯i	The incentive which used to purchase cloud storage space.	

The method of incentive collaborative data storage mechanism

This study assumes that for each data block, num/2−1 RSUs nodes are utilized for system storage. This approach not only guarantees a decentralized storage mechanism that strengthens the system’s robustness but also effectively minimizes data storage redundancy, optimizing storage efficiency.

In the study, a competitiveness matrix X is defined, where xij represents the competitiveness factor of node ni. The competitiveness factor is categorized into three types: storage capacity, distance, and driving record. The number of evaluation indicators n is set to 3 to reflect these three different aspects of competitiveness. To further optimize the node selection process, the driving record DR i is introduced to evaluate the historical behavior of the node and its mobility and stability within the network. The calculation formula for the driving record DR i is as follows: (1) DRi=a1∗11+VCi+a2∗ logMLi+1

where a1 and a2 reflect the influence of traffic violation records and driving mileage on the node selection, VCi represents the number of traffic violations, and MLi represents the driving mileage. (2) X=x1,1⋯x1,num⋮⋱⋮xn,1⋯xn.num==L1L2⋯LNMu1Mu2⋯MuNDR1DR2⋯DRN

where Li represents the path loss of the distance between node i and the sending node. The calculation formula is: (3) Li=l+10θlog10d/d0

where d0 is the reference distance, typically 1 meter; l is the path loss at the reference distance d0. θ is the path loss exponent, reflecting the complexity of the propagation environment; and d represents the distance between two points. Mui is the storage utilization, and the formula is (4) Mui=MiMi,max.

In addition, this article used the entropy weighting method to determine the weight of each evaluation index, and the weighting coefficient γn is defined as: (5) γn=1−Hnm−∑i=1mHn

(6) γ1+γ2+γ3=1

where γn represents the weight of the n-th competitiveness factor, such as physical distance, storage capacity, or driving records. The values are dynamically adjusted according to the current network state to ensure that the most important factors are prioritized in different scenarios. k is the adjustment coefficient, which controls the magnitude of the weight adjustment. A higher k indicates that the system is more sensitive to changes in competitiveness factors, while a lower k suggests that the system is more stable. fni represents the current state of the i th competitiveness factor, such as a node’s physical distance, storage capacity, or driving records, reflecting the node’s real-time performance in the network. (7) Hn=−k∗∑i=1N∑nifniln

(8) fni=xni ∑i=1Nxni

where k is a constant in the entropy weight method, defined as: (9) k=1lnN.

The final comprehensive competitiveness value ω_i is calculated using the following formula: (10) ωi=γ1Li+γ2∗Mui+γ3DR ^i

where L ˆi and DR ^i are derived from the normalization of the competitiveness factors of the storage node. (11) L ˆi=Li−LminLmax

(12) DR ^i=DRi−DRminDRmax.

Algorithm 1 Incentive mechanism collaborative data storage algorithm

	

In Algorithm 1, the selection process for collaborative storage nodes is demonstrated. In the first step, driving record DRi, node distance consumption Li, and storage utilization rate Mui are calculated using Eqs. (1), (3), and (4). The results are incorporated into the matrix in Eq. (2), yielding the competitiveness matrix. In the second step, the dynamic weight adjustment coefficients are calculated using Eqs. (5) through (9), and the competitiveness factors of the storage nodes are normalized using Eqs. (11) and (12). These are then substituted into Eq. (10) to obtain the comprehensive competitiveness value ωi for each storage node. In this study, competitiveness is equivalent to the incentive value obtained by the RSU node, with smaller competitiveness indicating higher priority. In the third step, the values of ω are sorted to identify the num/2−1 nodes with the highest priority as recipient nodes.

Figure 2 The system workflow diagram.

The proposed workflow, outlining the process where vehicles join the IoV network, collect and transmit data to RSUs, and how these RSUs manage data storage and offloading based on their competitiveness. It includes incentive allocation and data offloading to cloud storage when an RSU’s competitiveness is below a set threshold.

The method of incentive-based data offloading

In this study, the incentive allocation for nodes is based on three core factors: driving record, storage utilization, and path loss. The initial incentive value is calculated using the following formula: (13) Pi′=DRi∗β1−MiMi,max∗Li.

The driving record DRi evaluates the historical behavior of nodes, with well-performing nodes receiving more incentives. 1−MiMi,max the storage utilization reflects the remaining storage space; the more ample the storage space, the higher the incentive. The path loss Li represents the quality of signal transmission; the smaller the loss, the greater the incentive. Through the interaction of these factors, β can balance performance across various dimensions, ensuring fair and reasonable incentive distribution. (14) Pi=Pi′∗1−a∗Pi′Pmax.

In this formula, a is the attenuation coefficient, controlling the rate of incentive decay. As the incentive value Pi′ of a node approaches the threshold Pmax, the attenuation factor a∗Pi′ gradually increases, slowing down the rate of incentive growth. This mechanism ensures balanced incentive distribution, preventing excessive concentration of incentives on a few nodes, encouraging more nodes to participate in storage and data transmission, and maintaining system stability and fairness.

While providing incentives, collaborative storage needs to be implemented between nodes. Based on Eq. (15), the remaining storage capacity Mi of the node is obtained after collaborative storage. Here, M¯i represents the node’s remaining capacity before collaborative storage, and S is the size of the collaboratively stored data. (15) Mi=M¯i−S.

In this article, the threshold ratio T of competitiveness is defined in Eq. (15), l represents the number of times each RSU appears in the receiving node-set when traversing all RSUs. (16) T=lN.

In Eq. (10), as the remaining storage capacity of a node decreases, the required incentives for data storage increase. Eventually, when the incentives are insufficient to compete with other nodes, the node may no longer receive incentives, signaling the need for data offloading to alleviate storage pressure (Firdaus, Rahmadika & Rhee, 2021). (17) M ^i=Mi+r∗Mi,max.

Equation (16) outlines the process for calculating data offloading. Here, Mi ^ represents the remaining storage size in the node after the offloading process, whileMi denotes the initial remaining storage size before offloading. Additionally, r is the uninstallation ratio, and r∗Mi,max represents an integer multiple of the data block size unloaded from the node. This equation ensures that the unloaded data is a coherent block that aligns with the storage capacity and data block size of the system.

In Eq. (18), the remaining amount Moi of a roadside unit node depends on its original amount Mo¯i, the reward Pi obtained from receiving data blocks, the amount Sdbi spent by the node to broadcast data blocks and the amount P¯i used to purchase cloud storage space. (18) Moi=Mo¯i+Pi−Sdbi−P¯i.

Algorithm 2 Data Unloading Algorithm

	

In Algorithm 1, the first step involves calculating the incentive value granted to the collaborative storage nodes and the remaining storage capacities Pi and Mi of the nodes, using Eqs. (13), (14), and (15) based on the collaborative storage nodes obtained from Algorithm 1. In the second step, the algorithm traverses the nodes to calculate their competitiveness T. If a node’s T-value is below the lower threshold of competitiveness, data offloading commences. The offloading process continues, gradually increasing the offloading storage ratio until the node’s T-value reaches the upper threshold of competitiveness. In the third step, Eq. (18) is used to calculate the remaining incentive value of the node. To ensure efficient storage management, the amount of data offloaded must be an integer multiple of the data block size. The maximum amount of data that can be offloaded is equivalent to the total capacity of stored data. The competitiveness threshold limits were determined through testing, with an optimal range identified between 30% and 90%. Test results also indicate that as the number of blocks in the blockchain network increases, the frequency of offloads decreases, demonstrating optimal performance in terms of data management and network efficiency.

Experimental Evaluation

This section analyzes the collaborative data storage method from both security and experimental evaluation perspectives. The distributed storage model, combined with the immutability of blockchain, effectively ensures data integrity and reduces risks such as selfish nodes. The experimental analysis validates the system’s performance in terms of storage cost and scalability, and the simulation results confirm its effectiveness, along with the successful implementation of the incentive mechanism via smart contracts.

Experimental analysis

This section analyzes the performance of the proposed scheme in terms of storage cost and scalability. It then contrasts the scheme with existing work in the field. A comparative evaluation was conducted through simulation, which allowed for a controlled experimental environment to assess the performance metrics and compare them with those of other approaches.

The proposed system architecture was implemented using Python version 3.7.2, as the primary programming language. The smart contracts were developed in Solidity, a script-oriented programming language designed to implement smart contracts in the Ethereum blockchain. This article used Solidity version 0.5.17 for this purpose. Remix IDE, a web-based integrated development environment, was utilized for compiling, debugging, and testing these smart contracts. All experimental procedures were carried out on a single computer equipped with an Intel(R) Core (TM) i5-7300HQ CPU at 2.50 GHz and 16 GB of RAM, running the Windows 10 operating system. The underlying data conditions for the simulation are listed in Table 3.

Table 3 Experimental environment and basic data.

The underlying data conditions for the simulation are listed.

Experimental environment	Programming language	Python version 3.7.2.	
Smart contract	Solidity version 0.5.17	
Development tool	Remix IDE	
CPU.	i5-7300HQ CPU, 2.50 GHz	
RAM	16-GB	
Operating system	Windows 10	
Basic data	Number of RSUs	40	
Transaction size	0.25KB	
Block size	5KB (each block contains 20 transactions)	
Competitiveness threshold (higher limit)	90%	
Competitiveness threshold (lower limit)	30%	
The initial amount of wallet	150	

Storage consumption assessment

This study conducted two sets of experiments to compare storage consumption. In the first series of experiments, the storage consumption of the proposed scheme was benchmarked against that of the conventional blockchain, LayerChain, IoV block secure (G, Jatoth & Doriya, 2024) and GpDB schemes. In the second series, the study investigates how storage consumption is affected by varying the number of RSU nodes and data blocks, providing a comprehensive analysis of the scheme’s performance under different network conditions and data loads.

In the first set of experiments presented in Algorithm 2, this article observes that the remaining storage space of nodes in traditional blockchain configurations decreases as the number of data blocks increases. The results indicate that the proposed scheme (shown by red dashed lines and circles) demonstrates the lowest storage consumption, with a gradual increase as the block number grows. In contrast, the traditional blockchain shows a sharp rise in storage consumption, reaching the highest level at the 100-block mark. The LayerChain model exhibits a moderate increase in storage consumption, while the GpDB and IoV Block Secure models both show lower but steady increases in storage requirements. Overall, the proposed scheme proves to be the most efficient in terms of storage consumption, outperforming the other models as the block number increases.

These experiments varied the number of storage blocks from 10 to 100 and compared the storage consumption of our scheme with that of the traditional blockchain, GpDB, and LayerChain at equivalent block counts. The performance results indicate that our scheme exhibits significantly lower storage consumption owing to the incorporation of storage optimization and data offloading mechanisms under the incentivized framework.

In the second set of experiments depicted in Fig. 3, the study examined the storage consumption behavior in traditional blockchain networks. Because blockchain functions as a distributed ledger, each node must maintain a duplicate of the same data on the public chain. As the number of nodes increases, the total storage capacity required across all nodes escalates linearly for a constant number of data blocks. Similarly, for a fixed number of nodes, the total storage demand increases linearly with the number of data blocks.

Figure 3 Storage consumption comparison.

In the second set of experiments depicted in this figure, the study examined the storage consumption behavior in traditional blockchain networks. Because blockchain functions as a distributed ledger, each node must maintain a duplicate of the same data on the public chain. As the number of nodes increases, the total storage capacity required across all nodes escalates linearly for a constant number of data blocks. Similarly, for a fixed number of nodes, the total storage demand increases linearly with the number of data blocks.

This experiment is varied the number of RSUs from 10 to 70 and the number of data blocks from 50 to 150. Owing to the implementation of data offloading, the total storage consumption of RSU nodes was reduced as the amount of data shows that the overall storage consumption of the RSU nodes generally increased with the addition of more RSUs, despite instances of data offloading. When comparing the effects of increasing the number of blocks with a fixed number of RSUs, the total storage consumption typically increased with the block count, except in certain scenarios where data offloading led to a decrease in the total storage across all RSUs nodes.

The threshold value of competitiveness assessments

In this model, blockchain must address the issue of growing data within RSUs over time by optimizing storage through data offloading. Determining the circumstances under which to offload data and the extent of offloading are crucial for managing this challenge. This study uses the data’s competitive ability as the criterion for offloading. The lower limit of the competitiveness threshold is employed to decide when to initiate the offloading of data blocks, whereas the upper limit determines when to cease offloading.

The experiment presented in Fig. 4 compares the offloading folds for different data blocks by varying the number of data blocks from 50 to 175. Lower limits of 30% and 40% and upper limits of 70%, 80%, and 90% were tested. As observed in Fig. 4, the lower limit has a more significant impact on the number of offloads, whereas the upper limit has a relatively minor effect. The data revealed that the lowest average number of offloads occurred at a lower limit of 30% and an upper limit of 90%. Consequently, for the experimental tests in this study, the lower limit of the competitiveness threshold was uniformly set to 30%, and the upper limit was set to 90%.

Figure 4 Storage consumption for different number of RSUs.

The experiment presented in this figure compares the offloading folds for different data blocks by varying the number of data blocks from 50 to 175. Lower limits of 30% and 40% and upper limits of 70%, 80%, and 90% were tested. As observed in this figure, the lower limit has a more significant impact on the number of offloads, whereas the upper limit has a relatively minor effect. The data revealed that the lowest average number of offloads occurred at a lower limit of 30% and an upper limit of 90%. Consequently, for the experimental tests in this study, the lower limit of the competitiveness threshold was uniformly set to 30%, and the upper limit was set to 90%.

Furthermore, tests were conducted with the lower limit of the competitiveness threshold set to 20% to investigate the principle that a lower limit results in less offloading. When the lower limit of the competitiveness threshold was too low, the number of offloads from all the RSUs decreased, as did the amount of data offloaded each time. This eventually led to the depletion of the remaining storage in the individual nodes. Conversely, this issue did not arise when the lower competitiveness limit was set to 30%. Therefore, to ensure that RSU nodes always maintained some remaining storage capacity, the lower limit of the competitiveness threshold was established at 30%.

Data reception and offload assessment

This experiment comprises three parts: (1) The first part presents the total number of RSU offloads based on the variation in the number of RSUs. (2) The second part illustrates the change in the overall number of RSU offloads as the block size is altered. (3) The third part presents a comparison graph of the received and sent data for each RSU under the condition that the number of RSUs, block size, and block number remain constant.

In the first set of experiments presented in Fig. 5, the number of RSUs varied from 10 to 70, and the number of data blocks was changed from 50 to 150. The results indicate that the total number of data blocks offloaded by all RSUs increased as the number of RSUs increased while keeping the number of data blocks constant. The figure demonstrates that the increase in the number of RSUs did not affect the offloading process within individual RSU nodes, and multiple RSUs did not interfere with one another during the offloading process.

Figure 5 Upper and lower limit tests.

In the first set of experiments presented in this figure, the number of RSUs varied from 10 to 70, and the number of data blocks was changed from 50 to 150. The results indicate that the total number of data blocks offloaded by all RSUs increased as the number of RSUs increased while keeping the number of data blocks constant. The figure demonstrates that the increase in the number of RSUs did not affect the offloading process within individual RSU nodes, and multiple RSUs did not interfere with one another during the offloading process.

When the number of RSUs remained constant, an examination of the effects of increasing the number of data blocks revealed that the overall number of offloads for all RSUs continued to increase as the number of data blocks increased.

In the second set of experiments depicted in Fig. 6, the data block size was increased from 10 KB to 60 KB. For a constant number of data blocks, the overall total RSUs unload count increased as the data block size continued to increase. This indicates that the size of the data block affects the number of uninstalls.

Figure 6 Uninstallation times of different RSUs.

In the second set of experiments depicted in this figure, the data block size was increased from 10 KB to 60 KB. For a constant number of data blocks, the overall total RSUs unload count increased as the data block size continued to increase. This indicates that the size of the data block affects the number of uninstalls.

In the third set of experiments presented in Fig. 7, this study compared the received and sent data for each RSU under the condition that the number of RSUs, block size, and block number were fixed. The bar chart illustrates the storage occupied by the new data blocks (including the storage occupied by the data blocks generated by the node itself and the received data blocks) and the storage of the unloaded data blocks for each RSU node when the total number of storage blocks reaches 50. The line graph represents the ratio of received storage to unloaded storage for each RSU node when the total number of storage blocks reaches 50.

Figure 7 Variation of unloading times with different block sizes.

In the third set of experiments presented in this figure, this study compared the received and sent data for each RSU under the condition that the number of RSUs, block size, and block number were fixed. The bar chart illustrates the storage occupied by the new data blocks (including the storage occupied by the data blocks generated by the node itself and the received data blocks) and the storage of the unloaded data blocks for each RSU node when the total number of storage blocks reaches 50. The line graph represents the ratio of received storage to unloaded storage for each RSU node when the total number of storage blocks reaches 50.

As depicted in Fig. 8, the ratio remained stable between 1.1 and 1.5. Data blocks within the RSU node were organized in a queue structure and unloaded in a first-in, first-out sequence. Consequently, the size of the unloaded data blocks was smaller than or equal to that of the added data blocks. This ensured that the RSU node retained the most recently added data blocks after offloading, facilitating easy access to the latest data and enabling the node to efficiently manage temporary traffic issues. Moreover, every node actively participates in the storage and offloading of data, thereby ensuring that there are no selfish nodes in the system.

Figure 8 Comparison of storage consumption for each RSU.

As depicted in this figure, the ratio remained stable between 1.1 and 1.5. Data blocks within the RSU node were organized in a queue structure and unloaded in a first-in, first-out sequence. Consequently, the size of the unloaded data blocks was smaller than or equal to that of the added data blocks. This ensured that the RSU node retained the most recently added data blocks after offloading, facilitating easy access to the latest data and enabling the node to efficiently manage temporary traffic issues. Moreover, every node actively participates in the storage and offloading of data, thereby ensuring that there are no selfish nodes in the system.

Implementation of ethereum

This study presents an incentive-based collaborative data storage scheme implemented within an Ethereum smart contract. The implementation results are presented in two sections. The first part compares the amount of gas consumed to propagate a fixed number of data blocks within the network using the traditional blockchain. The second part compares the wallet balance of each RSU node after spending and receiving incentives to store and unload the data blocks within this scheme.

The first part is shown in Fig. 9. Transaction cost refers to the amount of gas consumed by smart contracts during transactions in the blockchain. Gas is the “fuel” in Ethereum that ensures the smooth operation of the Ethereum ecosystem and is the unit used to measure the computational work required for specific operations on the Ethereum blockchain. In this context, the cost of gas comprises two primary aspects: the cost required to store data for the receiving node and the cost required to offload data.

Figure 9 Comparison of transaction cost consumption.

The first part is shown in this figure. Transaction cost refers to the amount of gas consumed by smart contracts during transactions in the blockchain. Gas is the “fuel” in Ethereum that ensures the smooth operation of the Ethereum ecosystem and is the unit used to measure the computational work required for specific operations on the Ethereum blockchain. In this context, the cost of gas comprises two primary aspects: the cost required to store data for the receiving node and the cost required to offload data.

When comparing the cost of storing the same data block in this scheme with the traditional blockchain, it is evident that even though this scheme demands additional offload gas, the total cost remains one-third lower than that in the traditional blockchain.

In the second part of the results in Fig. 10, the number of data blocks was set to 30, 90, and 150. After transmitting the specified number of data blocks, the amount remaining for each RSU node after spending and receiving the incentives was calculated. The incentive changes in the figure demonstrate that all RSU nodes within the proposed framework actively participated in sending, receiving, and offloading data. The nodes were evenly distributed to ensure that there were no selfish nodes in the system.

Figure 10 Storage amount comparison.

In the second part of the results in this figure, the number of data blocks was set to 30, 90, and 150. After transmitting the specified number of data blocks, the amount remaining for each RSU node after spending and receiving the incentives was calculated. The incentive changes in the figure demonstrate that all RSU nodes within the proposed framework actively participated in sending, receiving, and offloading data. The nodes were evenly distributed to ensure that there were no selfish nodes in the system.

Discussion

Discussion on the validity of the framework

The dataset was randomly generated in the experiments to minimize dataset-related issues. The results compared data offloading times and storage consumption across varying data block size, block count, and RSU nodes. The analysis confirmed that the proposed incentive mechanism effectively encouraged cooperation among RSU nodes, leading to enhanced system efficiency. The proposed scheme outperformed traditional blockchain, LayerChain, and GpDB in storage consumption, saving over 99% in storage and reducing gas consumption by 35%, especially with more blocks. However, the scheme lacks a comprehensive study on data credibility, which is vital for ensuring network security and performance. Future work should address data verification to maintain trustworthiness in untrusted data scenarios.

Discussion on implementation

The framework relies on collaboration between blockchain and RSU nodes, optimizing storage and data transfer by adjusting storage and incentive strategies. The architecture includes modules for data block management, RSU incentives, and blockchain optimization. Experimental results showed improved storage efficiency and lower gas consumption compared to traditional blockchain technologies, even with more blocks. However, data credibility remains unresolved. Future research should focus on integrating data verification and encryption to enhance security and trustworthiness in real-world deployments.

Conclusions

This study introduces a collaborative data storage scheme with an incentive mechanism tailored for the BIoV ecosystem. This approach aims to enhance the security and scalability of the IoV while improving the storage utilization and data-sharing efficiency of RSUs. Initially, the scheme considers the distance between RSUs and the available storage space, selecting nodes for cooperative storage based on these factors. It then employs an incentive mechanism to encourage RSUs to receive data, thereby mitigating the influence of selfish nodes on collaborative storage efforts. Ultimately, when the level of competition reaches a certain lower limit, RSU nodes utilize the accumulated incentives to purchase cloud storage space, offloading data to the cloud layer to maintain adequate remaining storage space within the blockchain nodes. When compared to the traditional blockchain, GpDB, and LayerChain, this scheme demonstrated superior effectiveness in terms of storage consumption. This article implemented this scheme on the Ethereum platform by deploying smart contracts. The incentive comparison revealed that all nodes actively participated in data storage and offloading, with no selfish nodes.

Moving forward, future research will focus on introducing a certification mechanism to enhance the credibility of vehicle nodes, to improve overall trustworthiness within the IoV ecosystem. Specifically, we aim to explore the integration of federated learning for certifying node behavior and enhancing decision-making processes based on the trustworthiness of participating nodes. The objective is to ensure that only verified and trusted vehicle nodes participate in data storage and sharing. Moreover, practical applications of this certification mechanism could involve real-time monitoring and dynamic adjustment of node incentives to respond to changes in network conditions. These future advancements will contribute to securing and optimizing IoV systems, with measurable improvements in node reliability, data integrity, and overall network performance.

Supplemental Information

Supplemental Information 1 Code

Supplemental Information 2 Data

Supplemental Information 3 Supplemental Material

The authors would like to thank the PeerJ topic editor and reviewers for their constructive comments in the review process of this manuscript.

Additional Information and Declarations

Competing Interests

Author Contributions

Data Availability

The authors declare there are no competing interests.

Quan Shi conceived and designed the experiments, performed the experiments, analyzed the data, performed the computation work, prepared figures and/or tables, authored or reviewed drafts of the article, and approved the final draft.

Lankai Wang conceived and designed the experiments, performed the experiments, analyzed the data, performed the computation work, prepared figures and/or tables, authored or reviewed drafts of the article, and approved the final draft.

Chen Chen conceived and designed the experiments, performed the experiments, analyzed the data, performed the computation work, prepared figures and/or tables, authored or reviewed drafts of the article, and approved the final draft.

The following information was supplied regarding data availability:

The data and code are available at GitHub and Zenodo: https://github.com/wanglankai/CDSSIM-BIoV.

wanglankai. (2025). wanglankai/CDSSIM-BIoV: v1.0.0 (v1.0.0). Zenodo. https://doi.org/10.5281/zenodo.15123029.

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
