# Peer review of "A collaborative data storage with incentive mechanism for blockchain-based IoV"

_PeerJ Computer Science, doi:10.7717/peerj-cs.2849_

## Round 0.1 · original submission · Major Revisions

Carefully respond to the comments of the reviewers in a comprehensive revision

Reviewer 1 ·

Basic reporting

This paper presents A collaborative data storage by RSUs with incentive
mechanism for blockchain-based IoV, which are based on the all areas of IoT. Thus, this paper is directly related to the theme of this journal.
Overall, the paper is organized properly; So, the paper is accepted after following major changes:

1. Motivation and problem is not clear in introduction.
2. Heading 2 material methods must be renamed with Related work
3. Comments of Pseudo of algorithm is not given properly for understand of reader
4. Paper contains few grammar mistakes which will be cooperated in final version.
5. Only few references are used which are very less number of references. It’s better to increase references up to 40.
6. add few references related to IoT

Experimental design

experiment desing is fine

Validity of the findings

results are valid

Cite this review as

·

Basic reporting

Language and Clarity

The manuscript employs professional language and is mostly clear and unambiguous. However, some sentences are verbose and could benefit from simplification.

Examples:

Page 4: "The exponential growth of IoV data necessitates..." can be rephrased for brevity.

Page 5: "The experimental findings indicate..." is repetitive in its section.

Introduction and Background

The introduction effectively establishes the context of the research, emphasizing the challenges in IoV data storage and privacy.

The literature review covers relevant prior work, but additional discussion on emerging blockchain implementations in IoV contexts could enrich the background.

Figures and Tables

Figures and tables are well-labeled and relevant.

Some figures lack detailed captions, limiting their standalone interpretability. For instance, Figure 7 could provide more context about the experimental parameters.

Referencing

References are comprehensive and relevant, but there are inconsistencies in formatting (e.g., missing DOIs in some entries). Also the references are slightly old. It is suggested to add latest references in order to justify the contribution.

Experimental design

Originality and Scope

The study proposes an innovative framework combining RSU-based collaborative data storage and an incentive mechanism tailored for blockchain-based IoV systems.

The research is within the scope of the journal and addresses a critical gap in IoV data management.

Research Questions and Methods

The research questions are well-defined and relevant. However, the "knowledge gap" being addressed could be more explicitly stated in the introduction.

Methods are described in sufficient detail to allow replication, including algorithm descriptions (e.g., Incentive Mechanism Collaborative Data Storage Algorithm).

Ethical Considerations

Ethical considerations regarding data privacy and security are adequately discussed, given the blockchain context.

Validity of the findings

Data and Statistical Soundness

The data presented is robust and statistically sound. Comparisons with conventional methods (e.g., GPDB and LayerChain) are compelling.

The manuscript could provide additional sensitivity analysis to evaluate the impact of varying RSU densities on the proposed model's performance.

Conclusions

The conclusions are consistent with the results, emphasizing storage efficiency improvements and collaborative benefits.

Limitations, such as the lack of a comprehensive study on data credibility, are acknowledged. Future work suggestions align with these gaps.

Additional comments

Strengths:

The manuscript introduces a novel approach integrating incentive mechanisms into collaborative data storage for IoV.

Experimental results are comprehensive and demonstrate clear benefits over existing methods.

Weaknesses:

1) The language could be more concise in certain sections. Specifically, the subsections are starting directly below the main heading. It will be useful to introduce the section in a couple of lines before starting a subsection.

2) Figures and tables need enhanced descriptions.

3) Security and privacy are two very important aspects. There are no formal proofs of security or privacy that exist in the manuscript. It is suggested to add mathematical privacy and security gurantees in the paper.

Finally, the related work is not fully covered. For instance, a recently published paper on the same topic works very closesly to this manuscript. The link is provided below: https://link.springer.com/article/10.1007/s12083-024-01802-y
The author needs to compare their work to this closely related work. There is only one paper from 2024 and rest of the papers are slightly old to refer. It is suggested to add more latest papers for a better comparative analysis.

Reviewer 3 ·

Basic reporting

1. The abstract section requires revision. Avoid using phrases such as "we introduce".
2. Figure 6 should be described within the context of the Ethereum deployment practice to provide better clarity and relevance.
3. The legend in Figure 7 is missing the corresponding units.
4. The discussion of future research directions would benefit from greater specificity, including measurable objectives and potential practical applications.
5. The section "Formal Description and Analysis" lacks clarity and fails to adequately provide the theoretical foundation of the proposed method. It is recommended that the authors offer a more comprehensive conceptual explanation of the method, along with a detailed description of the fundamental processes. This would improve the reader's understanding and ensure the section effectively supports the overall contributions of the paper.
6. It is recommended that the authors conduct a thorough review of the paper’s English grammar. While the content is generally understandable, there are several grammatical errors and awkward phrasings throughout the text that may hinder clarity. Improving the overall language quality will enhance the
readability and professionalism of the paper. Consider revising sentence structures and correcting verb tense inconsistencies.

Experimental design

The authors have conducted a detailed analysis and experimentation of the proposed method, deploying and validating its effectiveness on Ethereum.

Validity of the findings

This paper introduces a novel collaborative data storage scheme enhanced by an incentivization mechanism and proposes two specialized algorithms: the Incentive Mechanism Collaborative Data Storage Algorithm (I-CDSA) and the Data Offloading Algorithm (DOA). The I-CDSA utilizes a competitiveness matrix to motivate receiving nodes to reduce storage consumption, thereby improving overall system efficiency. Meanwhile, the DOA leverages incentives to acquire additional cloud storage for efficient data offloading.

Cite this review as

---

## Round 0.2 · accepted · Accept

The authors have comprehensively addressed all reviewer comments from the previous round, substantially enhancing the overall quality of the manuscript. The revised version is now suitable for acceptance.

Reviewer 1 ·

Basic reporting

paper is revised according to comments and suggestions so I recommend as accept

Experimental design

experiment design are fine

Validity of the findings

results are valid

Additional comments

authors done good work to revised

Cite this review as

Reviewer 3 ·

Basic reporting

The editorial quality has been improved and I believe the paper's contribution warrants acceptance.

Experimental design

The editorial quality has been improved and I believe the paper's contribution warrants acceptance.

Validity of the findings

The editorial quality has been improved and I believe the paper's contribution warrants acceptance.

Additional comments

The editorial quality has been improved and I believe the paper's contribution warrants acceptance.

Cite this review as